# Learning to Imitate with Less: Efficient Individual Behavior Modeling in Chess

## Abstract

As humans seek to collaborate with, learn from, and better understand artificial intelligence systems, developing AI agents that can accurately emulate individual decision-making becomes increasingly important. Chess, with its long-standing role as a benchmark for AI research and its precise measurement of skill through chess ratings, provides an ideal environment for studying human-AI alignment. However, existing approaches to modeling human behavior require large amounts of data from each individual, making them impractical for new or sparsely represented users. In this work, we introduce Maia4All, a model designed to learn and adapt to individual decision-making styles efficiently, even with limited data. Maia4All achieves this by leveraging a two-stage fine-tuning method to bridge population and individual-level models and uses a meta-network to initialize and refine these embeddings with minimal data. Our experimental results show that Maia4All can accurately predict individual moves and profile behavioral patterns with high fidelity, establishing a new standard for personalized human-like AI behavior modeling in chess. Our work provides an example of how population AI systems can flexibly adapt to individual users using a prototype model as a bridge, which could lead to better and more accessible human-AI collaboration in other fields like education, healthcare, and strategic decision-making. Maia4All implementation is available at `https://anonymous.4open.science/r/UChess-3103`.

## 1 Introduction

The rise of artificial intelligence (AI) systems that rival or surpass human ability in domains where humans remain active has introduced the possibility of people collaborating with and learning from AI agents. A line of research has pursued this vision in the model system of chess, where AI became superhuman 20 years ago, people vary widely in their ability, and vast detailed datasets of action traces abound. Since capturing human decision-making style is a prerequisite to algorithmically-informed teaching and collaboration, previous work has focused on creating AI agents that mimic human play (McIlroy-Young et al., 2020; 2022; Jacob et al., 2022; Tang et al., 2024). Further, since capturing *individual* decision-making style is a prerequisite to personally tailored algorithmic instruction, researchers have developed models of how specific people play chess, surpassing population models in their accuracy rates on their target individual's decisions (McIlroy-Young et al., 2022).

However, these fine-tuning-based models require extraordinary amounts of data per person to function. When Maia, a human-like chess engine, was fine-tuned to play like specific individuals, gains in accuracy over base Maia were only achieved when the player had 5,000 games worth of data (McIlroy-Young et al., 2022). This is an immense amount of game-playing; a typical person would take around 1,000 hours to play this many games, which is equivalent to almost 25 weeks of full-time work at 40 hours per week. To put this in perspective, less than 1% of players on Lichess, a popular online chess platform, have played at least 5,000 games. Therefore, the fine-tuning-based approaches explored in previous work are proofs of concept that individual-level modeling is possible in chess, but it isn't a full solution to the problem because it doesn't work for the vast majority of people who would stand to benefit from algorithmically-informed teaching, learning, and collaboration.

How could we go about modeling individual-level decision-making behavior for people with much more modest amounts of data available? This is a difficult task for two reasons. First, existing models for modeling human decision-making, such as Maia and Maia-2 (McIlroy-Young et al., 2022;

Tang et al., 2024), are population models. This makes direct fine-tuning difficult, especially for low-resource players, as previously discussed. Second, human action prediction is formulated as a generative task to predict the next move that requires a model with strong generalization capabilities, which is particularly hard to achieve in a low-resource environment.

In this work, we propose **Maia4All**, a model that overcomes both of these challenges and can successfully predict chess moves at the individual level. Strikingly, **Maia4All** can model individual-level play with only 20 games of data. While Maia-2 shows virtually no progress when given 20 games of data played by a specific individual, and Maia fine-tuned with 1,000 games even gets worse, **Maia4All** significantly rises in accuracy from a baseline of 51.4% to 53.2%—a comparable rise to the accuracy gains reported in previous work using 5,000 games per player (McIlroy-Young et al., 2022).

We achieve data-efficient modeling of individual behavior in chess with two methodological contributions. First, we design a two-stage fine-tuning approach, where we first fine-tune Maia-2 to a diverse set of *prototype* players with rich game histories in order to adapt the model parameters from population-level modeling to individual-level modeling. Empirically, this makes it easier for the model to further adapt to low-resource players. In the second stage, this prototype-infused model with individual-level modeling capabilities is used as a bridge to be further fine-tuned with low-resource player data. Our second contribution is to start with a discriminative task instead of attempting the difficult generative task directly; we first find the most similar prototype player to the target player we want to model with a prototype-matching meta-network. Once we've identified a suitable prototype player, we initialize the target player's embedding with the prototype's embedding, and fine-tune on their limited data with this much better start.

Our framework not only provides state-of-the-art human modeling in chess, which can open the door to personalized AI teaching, learning, and collaboration, but it also holds potential for broader applications in domains where human-AI collaboration and algorithmically-informed education are possible.

## 2 RELATED WORK

**Human Behavior Modeling in Chess.** The challenge of creating a chess engine that can outplay any human was solved over 20 years ago. The research focus shifts towards extracting useful knowledge from these superhuman systems for humans. A direct way of doing this is to probe an AI chess engine in a human representation space. Evidence of human chess concepts learned by AlphaZero is found and measured by linear probes (McGrath et al., 2022). Going further, AlphaZero also encodes knowledge that extends beyond existing human knowledge but is ultimately learnable by humans (Schut et al., 2023). Another direction was the creation of a 'behavioral stylometry' model that can identify chess players from the moves they play (McIlroy-Young et al., 2021). Moreover, efforts have been made towards creating systems that can act as guides to humans (McIlroy-Young et al., 2020; Jacob et al., 2022; Tang et al., 2024), in which a model is trained to predict the next move a human will play, instead of optimizing for winning the game. In addition to predicting human actions at the population level, the models have been fine-tuned for individual-level human behavior modeling under *data-rich* settings (McIlroy-Young et al., 2022).

**Few-shot Learning and Meta Learning.** Few-shot learning focuses on the ability of models to learn and generalize from a very limited amount of labeled training data (Fei-Fei et al., 2006; Fink, 2004; Wang et al., 2020). Modeling unseen players follows the few-shot learning paradigm, where players' behavioral patterns are revealed by a limited collection of historical behaviors. Meta learning is a main approach to few-shot learning, aiming to improve novel tasks' performance by training on similar tasks. Meta learning can be categorized into metric-based methods (Vinyals et al., 2016; Snell et al., 2017; Koch et al., 2015; Sung et al., 2018) that aim to learn a similarity or distance function over objects and represent the relationship between inputs and the task space, model-based methods (Santoro et al., 2016; Munkhdalai & Yu, 2017), which focus on designing models with internal mechanisms to quickly adapt to new tasks, and optimization-based methods (Ravi & Larochelle, 2016; Finn et al., 2017; Nichol et al., 2018; Raghu et al., 2019) that aim to learn an initialization such that the model can adapt faster with few examples from there. Maia4All can be

regarded as a meta learning framework in that we learn a prototype matching meta network for player embedding initialization.

**Imitation Learning.** We can also frame this work as part of the imitation learning tradition, where a model is trained to perform some task after observing expert (human) demonstration(s) (Schaal, 1999; Zare et al., 2024; Wang et al., 2019). In the imitation learning context the model is usually attempting to learn a value function (inverse RL) (Ng et al., 2000), or to quickly learn an optimal solution to a given optimization problem (Schaal, 1999). In this paper we attempt to learn a *flawed* value function using *non-expert* demonstrations. Additionally, many imitation learning methods require the model to be in the same, or similar, state to the demonstrated one (Ho & Ermon, 2016; Zare et al., 2024) which is a condition that is impossible to guarantee in chess outside of the early game.

## 3 METHODOLOGY

### 3.1 OVERVIEW

As shown in Figure 1. (a) and (b), either the well-established human-like chess engine Maia (McIlroy-Young et al., 2020; 2022) and the state-of-the-art model Maia-2 (Tang et al., 2024) can be used as the population-level pre-trained foundation model to be fine-tuned towards individual players. We use Maia-2 as the base population model as it enables us to adjust its predictions by only varying the **population embeddings**. This property is particularly desirable for individual behavior modeling because the problem of adapting to individual players can be reduced to finding representative **player embeddings**. Thus the variant player embeddings can guide the personalized adjustments of human move predictions over a frozen model.

We face two challenges in fine-tuning Maia-2 for players with rare game histories. On the one hand, Maia-2 is a population model, which means its parameters are trained toward modeling common behavioral patterns among groups of players. This makes direct fine-tuning difficult, especially for low-resource players. Therefore, as shown in Figure1. (c), we design a two-stage fine-tuning approach to bridge population-oriented and individual-oriented parameters. To achieve this, we first fine-tune Maia-2 to a diverse set of *prototype* players with rich game histories (Maia-2-Prototype). In the second stage, the Maia-2-Prototype model with individual-level modeling capabilities is used as a bridge to be further fine-tuned to model low-resource players.

On the other hand, human action prediction is formulated as a generative task to predict the next move that requires a model with strong generalization capabilities, which is particularly hard to achieve in a low-resource environment. Therefore, we start with a discriminative task instead of attempting the difficult generative task directly. We first find the most similar prototype player to the target player we want to model with a prototype-matching meta-network, as shown in Figure 2. Once we've identified a suitable prototype player, we initialize the target player's embedding with the prototype's embedding, and fine-tune on their limited data with this much better start.

### 3.2 POPULATION MODEL

**Population Embeddings.** In chess, players can be meaningfully grouped by their skill level (McIlroy-Young et al., 2020; 2022), which is measured with now-pervasive rating systems that were originally developed for chess Elo (1967; 1978). Let $\mathbf{E}_P \in \mathbb{R}^{|\mathbf{E}_P| \times d}$ be the matrix of population embeddings, where each row corresponds to the embedding with dimension $d$ of a group of players that share a similar strength: $\mathbf{E}_P = [\mathbf{e}_{(0,1100]}, \mathbf{e}_{(1100,1200]}, ..., \mathbf{e}_{(2000,+\infty)}]^\top$. Given a player $i$ of strength level $r(i)$, we look up the embedding matrix $\mathbf{E}_P$ by rows to map the player strength to its embedding: $\mathbf{e}_i = \mathbf{E}_P[r(i)]$.

**Player Embeddings.** Similarly, given a set of individual players $I$, we denote their embedding matrix as $\mathbf{E}_I \in \mathbb{R}^{|\mathbf{E}_I| \times d}$, where each row corresponds to a player. Given a player $i \in I$, we look up $\mathbf{E}_I$ by rows to obtain the corresponding embedding of the active player: $\mathbf{e}_i = \mathbf{E}_I[i]$. For an unseen individual $u \notin I$, its embedding $\mathbf{e}_u \in \mathbb{R}^d$ will be initialized with prior knowledge and fine-tuned following the procedures in Section3.4.

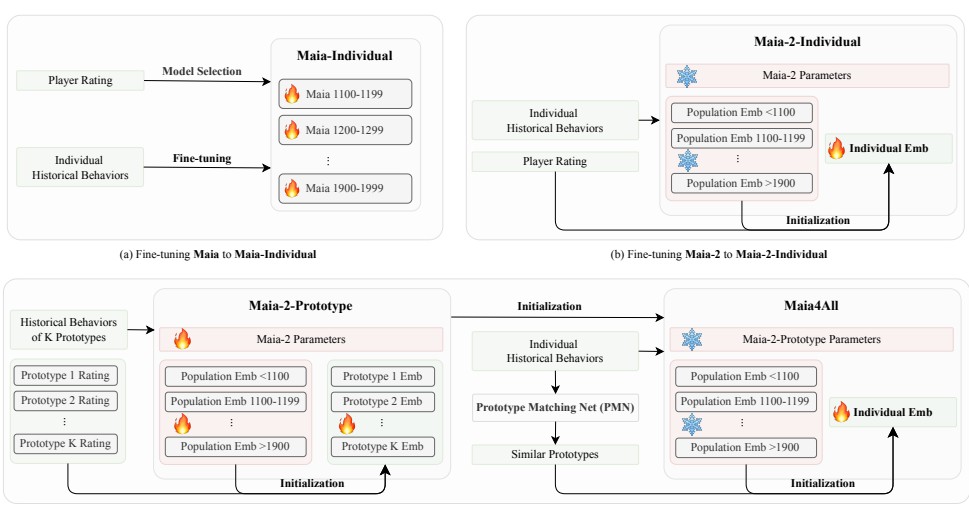

(a) Fine-tuning **Maia** to **Maia-Individual**

(b) Fine-tuning **Maia-2** to **Maia-2-Individual**

(c) Two-stage fine-tuning from **Maia-2** to **Maia4All** using **Maia-2-Prototype** as a bridge and **PMN** as a **Meta Network**

Figure 1: Overview of Maia4All training procedures.

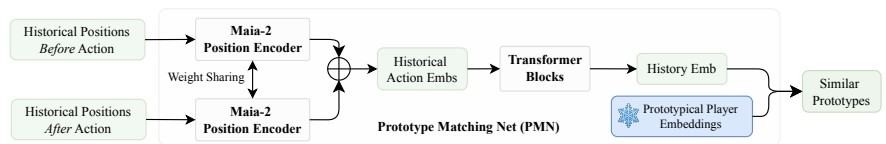

Figure 2: The architecture of the Prototype Matching Meta Network.

Maia-2 uses a ResNet (He et al., 2016) tower for chess position encoding and a skill-aware attention mechanism to bridge the population embedding and position embedding for human move prediction.

$$\textbf{Maia-2}(B, r(i)|\theta, \mathbf{E}_P), \tag{1}$$

where $B$ and $r(i)$ denote the encoded tensor representation of board positions and players' skill level as inputs, respectively, and $\theta$ represents the shared parameters across different groups of players (populations). $\mathbf{E}_P$ denotes the embedding matrix of populations.

### 3.3 MAIA-2-PROTOTYPE

We build on Maia-2 to develop the individual model:

$$\textbf{Maia-2-Prototype}(B, i|\theta', \mathbf{E}_I), \tag{2}$$

where $\mathbf{E}_I$ denote the embedding matrix of individual players. The shared parameters $\theta'$ are initialized with $\theta$. Since $\theta$ is already trained towards modeling diverse groups of players, it is easier to update to a universal set of parameters that models the more fine-grained individual-level move predictions.

While Maia-2-Prototype can be directly used to model the behaviors of individual players in $I$, it is unrealistic to include all players (e.g., 11 million on Lichess) in Maia-2-Prototype. On the one hand, learning a separate embedding for huge amounts of players can be challenging for computation and model learnability. On the other hand, most players haven't played enough games to support an accurately learned embedding for them. Furthermore, there are always new chess players, making it infeasible to have their embedding pre-trained. Therefore, we regard Maia-2-Prototype as a base model that can facilitate further adaptation to all individual players, in particular players with rare histories. This extensibility requirement guided us to select the player set $I$ in Maia-2-Prototype under two criteria: First, players should have sufficient historical games to ensure that their decision-making styles can be well learned in $\mathbf{E}_I$ and so $\theta'$ is not being interfered by under-trained embeddings.

Second, the number of players should be balanced for each strength level. This design not only helps the model to learn a universal set of parameters $\theta'$ for an unbiased distribution of players but also facilitates prototype-informed initialization.

Following previous work (McIlroy-Young et al., 2020), besides the main policy head to predict the human move, we include a value head to predict the game outcome as a regression task, where the labels 1, 0, -1 denote winning, drawing, and losing, respectively. To enhance the model's understanding of the game state, we inject auxiliary information as labels, including *legal moves* represented by multi-hot vectors and *human move information*: one-hot vectors of which piece is moved, which piece is captured (if any), the move's originating square, the move's destination square, and whether or not the move will deliver a check. It enriches the model's understanding of human chess moves by providing contextual insights beyond mere move indices and integrating essential objective knowledge of the game's mechanics, which has been proven to be helpful in human move prediction (Tang et al., 2024). This head is trained using bit-wise binary cross-entropy loss with multi-hot labels. The training objectives of these heads are balanced to contribute equally to model optimization.

## 3.4 MAIA4ALL

Since players can have limited historical data and there are always new players, it is crucial to ensure our proposed unified framework can effectively adapt to these players. We denote players that are not included in the training player set $I$ of Maia-2-Prototype as unseen players.

### 3.4.1 MAIA4ALL FINE-TUNING

Ideally, player strengths and styles can be well-learned by directly fine-tuning Maia4All with the new player embedding $\mathbf{e}$:

$$\textbf{Maia-2-Prototype}(B, u|\theta', \mathbf{e}_u) \xrightarrow{\text{Full fine-tuning}} \textbf{Maia4All}(B, u|\theta'', \mathbf{e}'_u) \tag{3}$$

where the full parameter set is fine-tuned. However, since modeling unseen players follows the few-shot learning paradigm, severe overfitting is expected when limited historical behaviors are provided. Moreover, full fine-tuning results in a shifted player embedding space for each player. As a result, the player embeddings learned are not transferable or directly comparable with other (pre-trained or future) player embeddings, which hinders its applications in downstream tasks such as behavioral stylometry (McIlroy-Young et al., 2021).

Since the shared parameters $\theta'$ in Maia4All are specifically learned towards modeling a wide range of individual players, this enables player embedding fine-tuning without updating $\theta'$:

$$\textbf{Maia-2-Prototype}(B, u|\theta', \mathbf{e}_u) \xrightarrow{\text{Emb fine-tuning}} \textbf{Maia4All}(B, u|\theta', \mathbf{e}'_u) \tag{4}$$

Note that the universal set of parameters $\theta'$ are shared across pre-trained or unseen players. The number of trainable parameters is drastically reduced to the embedding dimension $d$, preventing overfitting and enhancing knowledge sharing on the general understanding of individual behavior modeling.

Nevertheless, fine-tuning under low-resource settings is still challenging. Therefore, we propose to initialize unseen player embeddings with prior knowledge for data-efficient parameter updates. Such prior knowledge can be player strengths, as reflected by their ratings, and player styles, as derived from their historical behaviors.

### 3.4.2 STRENGTH INFORMED INITIALIZATION

In chess, decision-making style is informed by one's strength: e.g., a novice will typically not employ deep stratagems that indicative of grandmaster-level insight. Therefore, the player strength levels are ideal starting points to learn player styles:

$$\mathbf{e}_u \xleftarrow{\text{Initialize}} \mathbf{E}'_P[r(u)], \tag{5}$$

where $\mathbf{E}'_P$ denote the learned population embedding matrix in Maia-2-Prototype and $r(u)$ represent the strength level of player $u$.

However, player strengths are not accurately measured by the ratings until many games with different opponents are played and recorded. In real-world applications, player strength is very likely to be unknown and may be inaccurate. For example, when a grandmaster player joins an online chess platform as a new member, the rating will start to update from a preset rating that is close to the average ratings of the crowd. In this case, until many games are played on this platform, the player's rating will be far from the grandmaster level. Therefore, we need a procedure to initialize the player embedding that does not rely on the long-term strength measurements provided by ratings.

### 3.4.3 PROTOTYPE INFORMED INITIALIZATION

We aim to initialize unseen player embeddings with similar player embeddings. We denote the pre-trained players as *prototypes* to be matched. Since we balanced the number of players within each strength level during Maia-2-Prototype training, prototypical players should ideally cover the player styles within each level. We train a transformer-based meta-network for prototype matching, i.e., finding the most similar prototypical players. In particular, as shown in Figure 2, given a collection of historical behaviors of a prototype player, we use ResNet-based towers pre-trained by Maia-2 to extract positions before and after actions, and employ stacked Transformer layers to aggregate action embeddings. We use the frozen prototypical player embeddings $\mathbf{E}_I$ to measure the similarities and use cross-entropy loss for model training. The goal is to recognize the player given the historical actions. In the inference stage, we input the historical moves of *unseen* players to the meta-network and take the matched prototype embedding as the initialization for $\mathbf{e}_u$.

### 3.4.4 DISCUSSION

Note that both prototype matching and Maia4All exploit the same set of historical behaviors. However, prototype matching is a much easier task than human move prediction. This is because prototype matching is essentially a *discriminative* task against a fixed set of classes (prototypes), and human move prediction is a next-move *generative* task that requires a deeper understanding of the player's decision-making style. Therefore, we initialize the player embedding with the easier prototype matching task to get a rough understanding of how similar players behave and further calibrate the player embedding with human move prediction loss.

## 4 EXPERIMENTS

### 4.1 EXPERIMENTAL SETTINGS

**Datasets.** Online chess platforms feature a variety of game types, including blitz, rapid, and classical, each representing games played at different time controls (amount of time given to each player for the whole game). We use data from Lichess, a well-known large open-source chess platform, and its open database. In Lichess, since each game type is given a separate rating, ratings across different game types are not comparable (e.g. a rating of 1800 in "Rapid" is significantly weaker than a rating of 1800 in "Blitz" on Lichess). We focus on Blitz games because it's data-rich and we do not mix with other game types to ensure the ratings are meaningfully comparable. For individual model training and fine-tuning, we use the full game history in 2023 to compromise between the changing player strengths and styles over time and the availability of rich historical behaviors. We divide players into 11 bins: under 1100, over 2000, and nine 100-point wide strength bins from 1100 to 2000, i.e., $|\mathbf{E}_P| = 11$. During Maia-2-Prototype training, we use the game history of the $N$ most frequent players in each strength level, i.e., $|\mathbf{E}_I| = 11N$. We use 10 pre-trained and unseen players in each strength level for testing. We simulate unseen players with limited game history by limiting the training positions to the first $M$ positions, and we test them with the last 2048 positions in 2023. This yields the testing datasets for the testing datasets of prototypical and unseen players with 225,280 positions each.

**Implementation Details.** To maintain a consistent perspective from both sides of players, we used board flipping during training and testing; that is, positions with black to move were mirrored such that all analyses could be conducted from the white side's viewpoint. We further refined our dataset through game and position filtering, selecting only Blitz games with available clock information and disregarding the initial 10 plies of each game as well as positions where either player had less than 30

Table 1: Performance on unseen players under extremely low resource settings.

| | Move Prediction Accuracy | | | | Move Prediction Perplexity | | | |
|---|---|---|---|---|---|---|---|---|
| #Positions
#Games | 20000
≈500 | 8000
≈200 | 2000
≈50 | 800
≈20 | 20000
≈500 | 8000
≈200 | 2000
≈50 | 800
≈20 |
| Maia | 0.5132 | 0.5132 | 0.5132 | 0.5132 | 5.4530 | 5.4530 | 5.4530 | 5.4530 |
| Maia-2 | 0.5146 | 0.5146 | 0.5146 | 0.5146 | 4.5316 | 4.5316 | 4.5316 | 4.5316 |
| Maia-2-Individual | 0.5195 | 0.5196 | 0.5193 | 0.5189 | 4.4932 | 4.4939 | 4.4976 | 4.5022 |
| **Maia4All** | **0.5365** | **0.5348** | **0.5334** | **0.5322** | **4.2295** | **4.2431** | **4.2669** | **4.2988** |

Table 2: Performance on unseen players with 100,000 positions ≈ 2500 games.

| | Move Prediction Accuracy | | | | Move Prediction Perplexity | | | |
|---|---|---|---|---|---|---|---|---|
| | Skilled | Advanced | Master | Overall | Skilled | Advanced | Master | Overall |
| Maia | 0.4996 | 0.5099 | 0.5285 | 0.5132 | 5.8687 | 5.4642 | 5.1300 | 5.4530 |
| Maia-2 | 0.5008 | 0.5158 | 0.5364 | 0.5146 | 4.7900 | 4.4389 | 4.1936 | 4.5316 |
| Maia-2-Individual | 0.5071 | 0.5212 | 0.5400 | 0.5199 | 4.7264 | 4.4113 | 4.1760 | 4.4903 |
| **Maia4All** | **0.5261** | **0.5408** | **0.5554** | **0.5381** | **4.4018** | **4.1048** | **3.9219** | **4.1899** |

seconds remaining. The filtration is significant to eliminate the noise introduced by rushed decisions under time constraints, which could skew the true representation of a player's strength and style. We report all hyperparameters involved in training in Appendix Table 4.

**Evaluation Protocol.** We evaluate Maia4All with top-1 move-matching accuracy, which is essentially an extensive human study: we observe what humans would play in natural situations recorded by the Lichess Database, and see if it matches the predicted move of our system. We also measure the perplexity of move predictions, which reflects the model's confidence in its predictions. A lower perplexity indicates the model is more confident and accurate in human move prediction, as it corresponds to a higher likelihood (lower log-likelihood) of the correct human move. We report the results with three categories: *Skilled* (Blitz rating up to 1600, which slightly exceeds the initial rating of 1500), *Advanced* (Blitz rating between 1600 and 2000), and *Master* (Blitz rating over 2000, roughly comprising the top 10% of playersDuplessis).

**Baselines.** Maia (McIlroy-Young et al., 2020) is a set of 9 separate models, each trained on a different set of players at different skill levels from 1100 to 1900. Maia-1100 models the weaker players, Maia-1500 the intermediate players, and Maia-1900 the higher-skill players. We choose one of the Maia models for each population such that it is the nearest to their strength level for fair comparison. Since Maia-Individual (McIlroy-Young et al., 2022) is designed for data-rich settings, the published results of Maia-Individual indicate that it requires 5,000 games per player to show improvement over Maia. However, Maia4All, as a method for low-resource individual behavior modeling, at most has access to 100,000 positions (≈ 2,500 games). Therefore, we do not include Maia-Individual as a baseline. Since Maia-2 is trained towards adapting to diverse populations with a unified model, we use pre-trained $\mathbf{E}_P$ for the conditioning on different populations.

## 4.2 RESULTS

**Maia2.** As shown in Table 1 and Table 2, Maia-2 consistently outperforms Maia on both evaluation metrics under all settings. While top-1 accuracy gains are important, they may overshadow larger improvements in prediction quality. Such results show that Maia-2 can not only more accurately predict human behaviors but also be much more certain about its predictions. It is important to note that each Maia model is specifically trained for its respective strength level, relying solely on games where the active and opponent strength levels match in its training data. On the contrary, the unified modeling approach with player-aware attention in Maia-2 allows it to utilize a broader spectrum of

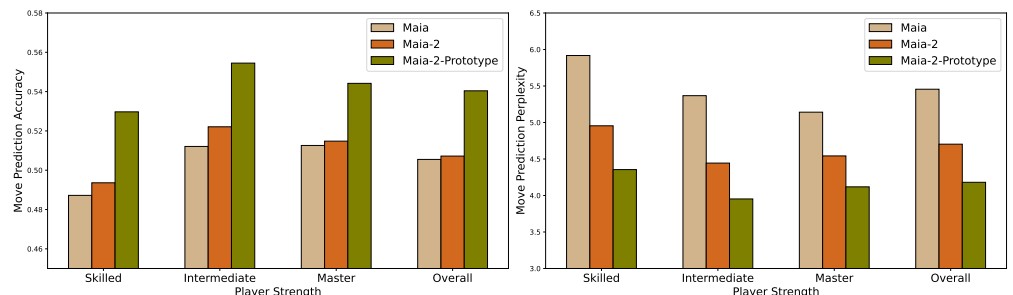

Figure 3: Performance on prototypical players with 100,000 positions ≈ 2500 games.

games, featuring a variety of strength level pairings, for training purposes. Such results support our design choice of choosing Maia-2 as the population model.

**Maia-2-Individual.** As shown in Table 1 and Table 2, directly fine-tuning from the population model (corresponds to Figure 1. (b)) barely improves human move prediction accuracy and perplexity under low-resource and relatively data-rich settings, which further motivate the two-stage fine-tuning procedure and the prototype-informed initialization.

**Maia-2-Prototype.** Previous work (McIlroy-Young et al., 2022) shows that population models can be fine-tuned to individual players, but require 5,000–10,000 games (worth around 200,000–400,000 positions) per player for performance to exceed the population model. At 1,000 games (worth around 40,000 positions) the fine-tuning was worse than the population model, likely due to over-fitting. As shown in Figure 3, Maia-2-Prototype outperforms both Maia and Maia-2 on both metrics with significant improvement. Moreover, we use at most 100,000 positions (around 2,500 games) per player for Maia-2-Prototype training, where 2,500 is somewhere between Maia can (1,000) and can not (5,000) show improvement by fine-tuneing. Such results demonstrate our proposed Maia-2-Prototype requires much less historical data to exhibit much stronger performance in modeling the decision-making style of individual players. The superior performance of Maia-2-Prototype also shows that the prototype player embeddings and the individual-oriented parameters are well-learned and ready to be used in the downstream Prototype Matching Net and prototype-informed initialization.

**Maia4All.** Similarly, we limit Maia4All to access 100,000 historical moves during fine-tuning to demonstrate the reduced size of historical data to achieve sufficient improvement. As shown in Table 2, Maia4All outperforms Maia with over 2 percentage points in accuracy and around 1.2 in perplexity (whereas Maia barely shows any improvement at this amount of data). These results demonstrate Maia4All's capability to adapt to unseen players with relatively rich data, and the amount of data needed is significantly lower.

When even fewer historical behaviors are accessible, Maia4All can still adapt to unseen players with considerable improvement in move prediction accuracy and perplexity. In particular, with only 800 positions (20 games, which is considered incredibly few for style modeling), Maia4All can transfer its predictions to unseen players with over 1.9 more percentage point and 1.1 lowered perplexity with prototype informed initialization. Note that the number of accessible positions is at most 20,000 position (worth 500 games) and Maia fine-tuning with 1,000 games is still showing negative improvement, which indicates fine-tuning Maia with our limited historical behaviors will result in even worse results than the original model.

**Prototype-Informed Initialization.** Strength-Init and Prototype-Init in Table 3 denote strength and prototype-informed initialization without further fine-tuning. Prototype-Init significantly performs better than Strength-Init on both metrics under all data scarcity settings. Strength-FT denotes the fine-tuned model based on Maia-2-Prototype with strength-informed initialization, and Maia4All can be regarded as Prototype-FT with the same naming strategy. The consistent superior performance of Maia4All over Strength-FT not only shows the effectiveness of the Prototype Matching Meta Net, but also supports that the better initialization provided by a discriminative task can be crucial to the final performance of the generative task.

Table 3: Performance of strength and prototype informed initialization.

| | Move Prediction Accuracy | | | | | Move Prediction Perplexity | | | | |
|---|---|---|---|---|---|---|---|---|---|---|
| #Positions
#Games | 100000
≈2500 | 20000
≈500 | 8000
≈200 | 2000
≈50 | 800
≈20 | 100000
≈2500 | 20000
≈500 | 8000
≈200 | 2000
≈50 | 800
≈20 |
| Strength-Init | 0.5008 | 0.5008 | 0.5008 | 0.5008 | 0.5008 | 4.8344 | 4.8344 | 4.8344 | 4.8344 | 4.8344 |
| Prototype-Init | 0.5180 | 0.5180 | 0.5175 | 0.5173 | 0.5167 | 4.5360 | 4.5360 | 4.5333 | 4.5400 | 4.5459 |
| Strength-FT | 0.5336 | 0.5308 | 0.5298 | 0.5279 | 0.5249 | 4.2599 | 4.3077 | 4.3238 | 4.3658 | 4.4151 |
| **Maia4All** | **0.5381** | **0.5365** | **0.5348** | **0.5334** | **0.5322** | **4.1899** | **4.2295** | **4.2431** | **4.2669** | **4.2988** |

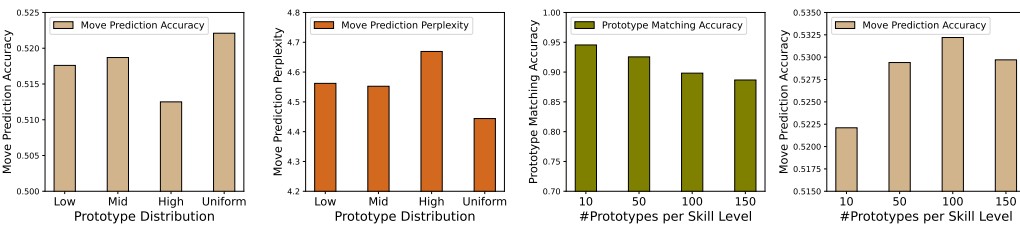

Figure 4: Effects of the distribution and number of prototypes to be matched.

**Prototype Matching.** The distribution of the prototypes to be matched is a hyperparameter. As shown in Figure 4, if we only include the prototypes from a biased distribution of the population, i.e, only select from low/medium/high-level players, it will result in lowered move prediction accuracy and raised perplexity compared to uniformly select $N$ prototypes from each strength level. Such results support our design choices of selecting the prototypes uniformly to cover the population space.

The number of prototypes $N$ per strength level is also a hyperparameter. Choosing an appropriate $N$ needs to compromise between the representativeness of prototypes for each range, i.e., more prototypes can better cover the player embedding space, and the difficulty in the prototype matching, i.e., more prototypes means more candidates to be classified against. This is evidenced by the results shown in Figure 4. We evaluate the top 1 matching accuracy of prototypical players under low-resource settings (800 positions). Increasing $N$ from 10 to 150 yields gradually lowered performance in prototype matching, while the best-performing Maia4All is achieved with a tradeoff between prototype matching accuracy and player embedding space coverage.

Note that the prototype matching network can be directly used for the behavioral stylometry (McIlroy-Young et al., 2021), i.e., identifying players given their historical behaviors. Since we freeze the shared parameters $\theta'$ and only finetune player embeddings for unseen players, the player embeddings are directly comparable. Therefore, our design supports behavioral stylometry off the shelf. With only 800 positions (around 20 games), our model can identify the player with 89% accuracy with 1 shot from 1100 candidates (100 players per strength level with 11 levels).

## 5  CONCLUSION

We introduce Maia4All, a model designed to learn and adapt to individual decision-making styles efficiently, even with limited data. Maia4All achieves this by leveraging a two-stage fine-tuning paradigm and using a meta-network to initialize and refine these embeddings with minimal data. Our experimental results show that Maia4All can accurately predict individual moves and profile behavioral patterns with high fidelity, establishing a new standard for personalized human-like AI behavior modeling in chess. Our work provides an example of how population AI systems can flexibly adapt to individual users using a prototype model as a bridge, which could lead to better and more accessible human-AI collaboration in other fields like education, healthcare, and strategic decision-making.

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

# A  REPRODUCTIBILITY

Table 4: Hyperparameter Settings.

| | |
|---|---|
| Initial learning rate | $1e^{-4}$ |
| Weight decay | $1e^{-5}$ |
| Batch size (positions) | 8192 |
| Minimum move ply | 10 |
| Maximum move ply | 300 |
| Remaining seconds threshold | 30 |
| #Backbone blocks $K_{Conv}$ | 12 |
| #Attention block $K_{Att}$ | 2 |
| #Input channels $C_{\text{input}}$ | 18 |
| #Intermediate channels $C_{\text{mid}}$ | 256 |
| #Encoded channels $C_{\text{patch}}$ | 8 |
| Player embedding dimension $d$ | 128 |
| Attention head dimension $d_h$ | 64 |
| Attention intermediate dimension $d_{\text{att}}$ | 1024 |
| #Attention heads $h$ | 16 |
| player per range $N$ | 100 |

## A.1  POSITION REPRESENTATION AND ENCODING

We follow the well-established prior works McIlroy-Young et al. (2020); Silver et al. (2017) to represent chess positions as multi-channel $8 \times 8$ matrices, including:

- Piece Representation: The first 12 channels categorize the board's pieces by type and color, with one channel each for white and black Pawns, Knights, Bishops, Rooks, Queens, and Kings. A cell is marked 1 to denote the presence of a piece in the corresponding location, and 0 otherwise.
- Player's Turn: A single channel (the 13th) indicates the current player's turn, filled entirely with 1s for white and 0s for black, providing the model with context on whose move is being evaluated.
- Castling Rights: Four channels (14th to 17th) encode the castling rights for both players, with the entire channel set to 1 if the right is available or 0 otherwise.
- En Passant Target: The final channel (18th) marks the square available for en passant capture, if any, with 1 and 0s elsewhere.

One important departure from previous work is that we only use the current chess position, and not the last few chess positions that occurred in the game (models have typically incorporated the six most recent positions in the game). Many games with perfect information, including chess, can be modeled as alternating Markov games Littman (1994); Silver et al. (2016), where future states are independent of past states given the current game state. Therefore, the current chess position theoretically encapsulates all the information necessary to make future decisions. Although human decision-making in chess may sometimes subtly depend on the historical lead-up to the current position, these effects are anecdotally small.

In exchange, we gain two large practical benefits. First, modeling AI-human move matching in a Markovian way vastly improves training *efficiency* by reducing the computational load via significantly smaller data usage for each decision. Second, it also enhances *flexibility*, enabling our resulting model to make predictions even without historical data, which is particularly advantageous in situations where only the current position is available, like chess training puzzles or any position that didn't necessarily occur in a full game.

