# OpenReview forum: "Learning to Imitate with Less: Efficient Individual Behavior Modeling in Chess"
_ICLR.cc/2025/Conference — Submitted to ICLR 2025_

### Official Review · Reviewer_BJxq · 2024-10-28

**Soundness:** 3
**Presentation:** 3
**Contribution:** 3
**Rating:** 5
**Confidence:** 3

**Summary:**

The paper introduces Maia4All, a model that efficiently learns and adapts to individual chess players' decision-making styles with limited data. The authors develop a two-stage fine-tuning process: they first adapt a population-level model, Maia-2, to prototype players with rich game histories; the model then uses a meta-network for prototype-informed initialization to adapt efficiently to individual players with as few as 20 games. Experimental results show that Maia4All achieves high accuracy in move prediction. Overall, the work provides an example of how population AI systems can flexibly adapt to individual users using a prototype model as a bridge in the field of chess.

**Strengths:**

- The paper introduces a novel application of few-shot learning in chess. The two-stage fine-tuning approach is particularly innovative, allowing the model to leverage rich data from prototype players while efficiently adapting to low-data individuals.
- The experimental design is rigorous, with clear evaluation metrics such as move prediction accuracy and perplexity across various data scarcity settings.
- Very well written and easy to follow: The paper includes step-by-step explanations and visual aids that clarify the workflow of the proposed model.

**Weaknesses:**

- This paper exclusively studies behavior modeling in the context of chess, raising concerns about whether the methods proposed can generalize effectively to other domains. While the results show that the authors have clearly made an improvement in behavior modeling in chess -- especially when player data is low -- the authors haven't thorough evidence that these methods can improve behavior modeling in other domains which limits the impact of this work. Concretely the authors say "Our work provides an example of how population AI systems can flexibly adapt to individual users using a prototype model as a bridge," which I agree with but follow with "which could lead to better and more accessible human-AI collaboration in other fields like education, healthcare, and strategic decision-making." which I am not convinced of.

- The method could also benefit from further ablation studies to clarify the relative impact of each component of the proposed method/system.

**Questions:**

1. Can this method be applied to other domains outside of chess?
2. How would we model human behavior for domains where human behavior may be less structured or less data-rich.

---

### Official Review · Reviewer_i5fP · 2024-10-31

**Soundness:** 2
**Presentation:** 1
**Contribution:** 2
**Rating:** 3
**Confidence:** 3

**Summary:**

This paper addresses the challenge of modeling individual behavior in chess, particularly for players with limited historical game data. It introduces Maia4All, trained using a two-stage fine-tuning approach. In the first stage, the base population model is fine-tuned on a diverse set of prototype players with extensive game histories, serving as a bridge to further fine-tune on low-resource player data in the second stage. A prototype-matching meta-network is used to identify the prototype player most similar to the target player，which is essential for efficient fine-tuning with limited data. Experimental results show that Maia4All improves in predicting individual moves and profiling behavioral patterns, even with as few as 20 games of data.

**Strengths:**

This paper addresses the challenge of behavior modeling in a realistic setting with limited player game records. Existing methods for modeling human behavior require extensive data from each individual, limiting their practical use. The motivation and areas for improvements are clearly outlined.

**Weaknesses:**

1. Overall, I find certain aspects of the methodology presentation unclear. The study builds on Maia-2, but several settings from Maia-2 need further explanation. For example, in Section 3.1, population embeddings are introduced, but there is no explanation of how Maia-2's predictions are adjusted solely by varying these embeddings. A brief overview of how Maia-2 works with population embeddings would be helpful. Additionally, in Section 3.2, more detail is needed on the population embeddings and player embeddings, specifically how these matrices are generated and adjusted.


2. This method builds on Maia-2 improvements and enables behavior modeling for new players with less data; however, within this framework, training is still required for each player, making the process time-consuming and less scalable for larger populations. The framework’s overall level of innovation is somewhat limited. I was expecting more discussions about how to address the scalability issue and make the per-player training more efficient.

**Questions:**

1. As in mentioned in weaknesses, how to get population embedding matrix and individual embedding matrix needs more explanations.

2. In 3.4.1, how to finetune player embedding  without updating $ θ' $?

3. The presentation of the meta-network for prototype matching in Figure 3 is unclear. It would be helpful to provide more specific details on how this structure uses History Embeddings to match similar Prototypical Player Embeddings.

4. In Section 3.4.2, prototypical player embeddings are compared with aggregated action embeddings to find the most similar prototypical players. I was expecting the authors to provide quantitative results on the prototype matching accuracy.

---

### Official Review · Reviewer_3nqy · 2024-11-04

**Soundness:** 3
**Presentation:** 2
**Contribution:** 1
**Rating:** 3
**Confidence:** 4

**Summary:**

This paper is about improving the ability of a pre-trained chess neural network (Maia-2) to match individual behavior in chess with little data (20 games). The overall approach involves first fine-tuning the base model on a set of "prototype" players balanced across strength levels, and the fine-tuning embeddings initialized from the most similar prototype players. The results show marginal improvements in matching individual chess behavior with few games (~51% -> ~53%).

**Strengths:**

- I have not seen the idea of matching embeddings of (low-resource) players with other prototypical players, which is quite interesting.
- I like the ablation of strength and prototype-initialization without fine-tuning.

**Weaknesses:**

- The paper is not sufficiently well-written. There are many things unclear to me as a reader. For example:
    - How does the model use the population/individual embeddings?
    - Since you want to predict next moves, why do you need a value head and auxiliary losses?
    - It is difficult for me to understand what’s going on in Figure 1.
- A more fundamental issue with the work is its generality. This paper proposes an initialization technique to solve a problem that is very specific (low-resource individual behavior modeling in chess). It is unclear to me if such techniques could be useful for other more practical tasks (e.g., learning the writing style of a person with 20 documents).
- The proposed method, given its complexity, provides marginal improvements over no fine-tuning baselines at best. A 2% improvement in move-matching over the baseline means the model predicts an additional move correctly *in every 50 moves*! Is this even meaningful?
- Overall, I'm slightly suspicious that any fine-tuning is required after prototype matching. Chess is such a closed domain that there should really be quite a small number of "styles" or prototypes for each skill level. With a sufficient number of prototypical players and a good prototype classifier, you should do really well with prototype matching alone. One experiment I would suggest is assuming access to an "oracle" prototype classifier that always outputs the best prototype player, and compute the max move prediction performance among all prototype players.

**Questions:**

1. Section 3.1: Where are the population embeddings from? If this is background information about Maia-2, you should introduce it.
2. Not sure what this means: “Since θ is already trained towards modeling diverse groups of players, it is easier to update to a universal set of parameters that models the more fine-grained individual-level move predictions.

---

### Meta-Review · Area_Chair_CsX7 · 2024-12-21

**Metareview:**

While the paper presents an interesting approach to individual behavior modeling in chess, the combination of marginal improvements, limited generalizability, and unclear technical details suggests the work needs substantial revision before it meets the conference standards.

**Additional Comments On Reviewer Discussion:**

There were no rebuttals, and the reviews seem consistent.

---

### Decision · Program_Chairs · 2025-01-22

Reject